# Formation of Self-Assembled Liquid Crystalline Nanoparticles and Absorption Enhancement of Ω-3s by Phospholipids and Oleic Acids

**DOI:** 10.3390/pharmaceutics14010068

**Published:** 2021-12-28

**Authors:** Sang-Won Jeon, Han-Sol Jin, Young-Joon Park

**Affiliations:** 1College of Pharmacy, Ajou University, Worldcup-ro 206, Yeongtong-gu, Suwon-si 16499, Korea; jsw0603@imdpharm.co.kr; 2Research Center, IMDpharm Inc., 17 Daehak 4-ro, Yeongtong-gu, Suwon-si 16226, Korea; hansol@imdpharm.co.kr

**Keywords:** Ω-3 fatty acids, phospholipids, oleic acid, liquid crystalline nanoparticles

## Abstract

This study aimed to optimize and evaluate self-assembled liquid crystalline nanoparticles (SALCs) prepared from phospholipids and oleic acid for enhancing the absorption of Ω-3s. We explored the structure and optimal formulation of SALCs, which are composed of Ω-3 ethyl ester (Ω-3 EE), phospholipids, and oleic acid, using a ternary diagram and evaluated the improvement in Ω-3 dissolution, permeation, and oral bioavailability. The in vitro dissolution and pharmacokinetics of Ω-3 SALCs were compared with those of Omacor soft capsules (as the reference). The shape of the liquid crystal was determined according to the composition of phospholipids, oleic acids, and Ω-3s and was found to be in cubic, lamellar, and hexagonal forms. The dissolution rates of eicosapentaenoic acid (EPA) and docosahexaenoic acid (DHA) obtained from Ω-3 SALCs were 1.7 to 2.3-fold higher than those of the Omacor soft capsules. Furthermore, a pharmacokinetic study in male beagle dogs revealed that Ω-3 SALCs increased the oral bioavailability of Ω-3 EE by 2.5-fold for EPA and 3.1-fold for DHA compared with the reference. We found an optimal formulation that spontaneously forms liquid crystal-based nanoparticles, improving the bioavailability of EPA and DHA, not found in the existing literature. Our findings offer insight into the impact of nanoparticle phase on the oral delivery of oil-soluble drugs and provide a novel Ω-3 EE formulation that improves the bioavailability of EPA and DHA.

## 1. Introduction

The Ω-3s, also known as *n*-3 polyunsaturated fatty acids (PUFAs), are essential fatty acids and important cell membrane components [1]. The consumption of Ω-3s, such as alpha-linolenic acid (ALA), eicosapentaenoic acid (EPA), and docosahexaenoic acid (DHA), is considered critical for public health [2]. The global cardiovascular disease problem arises from cardiometabolic risk factors, such as hyperglycemia, hypertension, obesity, hypercoagulation, inflammation, and dyslipidemia [3,4,5,6]. The administration of Ω-3 fatty acids is a remedial treatment for cardiovascular diseases and depressive disorders in adults [3,4,5,6,7]. Moreover, Ω-3 fatty acids reduce oxidative stress and muscle damage and inhibit thrombosis and platelet aggregation [8,9,10].

The Ω-3 ethyl esters (Ω-3 EE) of Ω-3 fatty acids are used as a prescription drug approved by the FDA to treat hypertriglyceridemia in most cardiovascular diseases and have a strong safety profile when taken as directed. Nevertheless, during administration, Ω-3 fatty acids are unstable and insoluble in water [11]. The long-chain fatty acids EPA and DHA have been found to be absorbed by the gastrointestinal (GI) tract in their free form effectively [12,13,14,15,16]; their original molecules enter the micelles independently and are taken up by enterocytes through diffusion or transport systems along with CD36/FABP (fatty acid-binding protein) or FATP4 (fatty acid transport protein 4) [17,18,19]. Low bioavailability and limited oral absorption of Ω-3s are the factors responsible for their low dissolution in the GI tract and the limit of diffusion of the unstirred water layer, which has been considered a rate-limiting step in the intestinal uptake of drugs with low solubility and high permeability, such as oil-soluble drugs [20,21,22]. Therefore, the benefits of Ω-3 fatty acid supplements are not fully realized due to their poor solubility and bioavailability [23].

The Ω-3s are bioactive lipids with a three-dimensional supramolecular structure. A component of Ω-3s, EPA, forms a liquid crystalline structure when mixed with the lyotropic non-lamellar lipid monoolein, leading to the establishment of compartmental bodies. Ω-3 PUFA nanosponges may serve as mesoporous reservoirs, allowing for the encapsulation of EPA with hydrophobic drugs and hydrophilic macromolecules to promote the targeted treatment of neurodegenerative and other diseases [11,24].

Drug formulations, such as oil-in-water emulsions and EPA micelles with the lyotropic non-lamellar lipid monoolein, form a multicompartment assembly of lyotropic liquid crystalline nanoparticles that are utilized for delivering Ω-3 fatty acids [11,25,26,27,28,29,30]. Recently, l-lysine salts of carboxylic EPA and DHA have also been developed. Jing and Lin et al. reported that free fatty acids (FFA) exhibited fast absorption and high bioavailability [12]. Moreover, multiple doses of Ω-3 carboxylic acids (Ω-3 CA) in a Chinese population have been used to estimate pharmacokinetics, safety, and tolerability profile of Ω-3 CA in healthy subjects. The exposure levels of EPA and DHA in the blood were consistent with those in other indigenous populations [31].

A previous study reported that the absorption of EPA, DHA, and EPA plus DHA was constant, with median *T_max_* occurring at approximately 5.5–6 h following single and multiple oral Ω-3 CA, 4 g doses; a disadvantage of this study was that 42.9% of the participants experienced an adverse effect (diarrhea) during Ω-3 CA treatment [31].

Strategies to improve the aqueous solubility and bioavailability of Ω-3 fatty acids in the GI tract include dosing regimens with the concomitant intake of a high-fat diet [32,33], chemical modification (e.g., re-esterified TG, free fatty acids, and phospholipid forms) [33,34,35,36,37], and pharmaceutical formulations [23,38,39,40,41]. Although several studies have attempted to improve the absorption of Ω-3s, these fatty acids still have poor absorption (particularly in a fasted state), an expensive manufacturing process, and stability issues [37,42].

Drug delivery systems are used to improve the solubility and increase the bioavailability of poorly water-soluble drugs [43,44,45,46,47,48,49,50,51,52,53]. Numerous techniques have been developed in recent years to enhance the oral bioavailability of poorly water-soluble drugs, including mesoporous silica, microemulsions [54], self-nano-emulsifying drug delivery systems [55], lyotropic liquid crystalline nanoparticles, and chitosan nanoparticles [56]. However, lyotropic liquid crystalline nanoparticles as drug delivery systems are associated with some concerns over their specific structural tendencies [57,58]. Nonetheless, Luzzati et al. reported the self-assembly of liquid crystalline nanoparticles (SALCs) formed by an amphiphilic liquid for the first time, which may offer several advantages, including increased solubility, bioavailability, high thermodynamic stability, and controlled release [53,59,60,61,62,63]. The most interesting SALC phases are hexosomes and cubosomes with internal hexagonal and cubic structures, respectively [64,65]. From the perspective of various liquid crystal phases, the molecular structure of amphiphilic lipids, a component of SALCs, is the most important factor along with concentration, pH, and temperature [66,67,68,69]. The particle-carrying hexosomes are stabilized using the copolymer F127 and an emulsifier. These mechanisms aim at achieving a stabilization effect and temperature control of the internal nanostructure. Accordingly, highly ordered nanocarriers have been developed using nanoparticles that deliver drugs with beneficial health effects [70].

The present study aimed to prepare and optimize SALCs, based on their structural characteristics, using phospholipids and oleic acid, and investigate the effect of the molecular structure of lipids on the liquid crystal phase behavior in the GI tract and the absorption of Ω-3 fatty acids in male beagle dogs.

The novel approach presented in this study for producing self-assembled liquid crystalline nanoparticles represents one of the attempts to improve patient compliance when administered excess excipients and increase the oral absorption of oil-soluble drugs, such as Ω-3s. Here, we present a solubilization technology for the oral delivery of Ω-3 fatty acids with effective absorption and improved clinical use.

## 2. Materials and Methods

### 2.1. Materials

Ω-3 EE were purchased from K.D. Pharma GmbH (Bexbach, Germany). Oleic acid (OA purity 88.5%) was obtained from Croda (New Castle, DE, USA). Phospholipids (PC, phosphatidylcholine 75%, lecithin) were purchased from Lipoid (Ludwigshafen, Germany). Omacor (Ω-3 EE) soft capsules were purchased from Kuhill Pharm (Cheonan-si, Korea). FaSSIF/FeSSIF/FaSSGF powders were purchased from Biorelevant (London, UK). Sterile syringe PTFE filters (0.45 µm, 25 mm) were purchased from Pall Corporation (Port Washington, NY, USA). Tris (hydroxymethyl)-aminomethane ≥ 99.0% and phospholipase from *Thermomyces lanuginosus* were purchased from Sigma-Aldrich (St Louis, MO, USA). Sodium chloride (99.0%) and calcium chloride dehydrate (71.0–77.5%) were purchased from Samchun Pure Chemicals (Pyeongtaek-si, Korea). Acetonitrile for HPLC (isocratic grade) was purchased from Merck Millipore (Billerica, MA, USA). All chemicals used were of analytical grade and obtained from Samchun Pure Chemicals (Pyeongtaek-si, Korea).

### 2.2. Ternary Phase Diagram

To determine the composition of Ω-3 SALC formulations, a ternary phase diagram evaluation was performed with Ω-3, PC, and OA, and the properties of spontaneous liquid and liquid crystals were confirmed. Briefly, Ω-3 SALC formulations were prepared by mixing Ω-3 EE, OA, and PC in specific ratios. Ω-3 EE and OA were first mixed in the ratios of 1:9, 2:8, 3:7, 4:6, 5:5, 6:4, 7:3, 8:2, and 9:1 (*w*/*w*). For the ternary phase diagram, the oil mixture and PC were mixed in the range of 1:9 to 9:1 (*w*/*w*). Next, the Ω-3 formulations were homogeneously mixed, using a magnetic stirrer and a hand-held homogenizer (Polytron PT 1200E; Kinematica AG, Malters, Switzerland), for 5 min. When Ω-3 formulations reached thermal equilibrium at 20–25 °C, they were mixed with fed-state simulated intestinal fluid (FeSSIF) and evaluated, using polarized optical microscopy (S38; Microscopes Inc., St Louis, MO, USA), to determine the ternary phase diagram and define the lamellar, cubic, and hexagonal phases. FeSSIF was prepared by dissolving FaSSIF/FeSSIF/FaSSGF powders (11.2 mg/mL) in blank FeSSIF (4.04 mg/mL NaOH, 8.65 mg/mL glacial acetic acid, and 11.87 mg/mL NaCl—adjusted to pH 5.0) [71].

### 2.3. Preparation of SALC Formulations

In the ternary phase diagram, six formulations (F1–F6) of SALCs with different liquid phases (lamellar, hexagonal, cubic, and non-lipid phases) were selected, as shown in Table 1. First, Ω-3 EE, OA, and PC were mixed in different ratios. The mixture was then ground homogeneously under nitrogen gas for 5 min. In this study, the formulations were designed for SALCs based on phospholipids to enhance the absorption of Ω-3 EE.

### 2.4. Characterization of Formulation

#### 2.4.1. Particle Size Analysis

Particle size measurement of SALCs was performed by dispersing the Ω-3 formulations in FeSSIF (pH 5.0) at 37 °C for 1 h and filtering through a PTFE membrane (Pall Corporation). A dynamic laser scattering (DLS) system (NICOMP 380ZLS; Particle Sizing Systems, Port Richey, FL, USA) was used to measure the particle size of SALCs. In a borosilicate glass disposable tube, 6 mm × 50 mm samples were measured at a setting temperature of 120 °C, index of refraction of 1.333, channel width of 15 µsec, liquid viscosity of 0.933 cP, and scattering angle of 90°. The particle size distribution was calculated in the ‘intensity’ analysis configuration [11].

#### 2.4.2. Morphology

The phase behavior of SALCs was studied using polarized optical microscopy. To observe the phases of SALCs, 100 µL of Ω-3 formulations was placed on a glass slide and mixed with 1–2 drops of FeSSIF. After a 10 min incubation at 20–25 °C, the cover glass was placed on the glass slide and visualization was conducted at 40× and 100× magnification. A field emission transmission electron microscope (FE-TEM) (Tecnai G2 F30 S-twin; FEI, Hillsboro, OR, USA) was used to identify the inner structure of the SALCs [72]. For FE-TEM studies, the preparative procedure for DLS measurement was used for sample preparation. First, the Ω-3 formulations were dispersed in FeSSIF, incubated for 1 h in a water bath maintained at 37 °C, and filtered through a PTFE filter to obtain the SALCs. Thereafter, 10 µL colloidal solutions of SALCs were placed on 300-mesh carbon-coated copper grids (Ted Pella Inc., Redding, CA, USA) and stained with a 1% (*w*/*v*) phosphotungstic acid solution; then, the excess fluid was removed by drying in a desiccator overnight at 20–25 °C. The samples were subsequently examined at 300 kV.

10 µL of self-assembled LCNPs was placed on the surface of a silicon wafer and left to dry to characterize the morphology of LCNPs using atomic force microscopy (AFM) (XE-150; PSIA, Suwon-si, Korea). The sample was scanned in an area of 2 μm × 2 μm in the non-contact mode at a scan frequency of 0.5 Hz.

### 2.5. Dissolution

The in vitro dissolution test was performed using USP apparatus 2 (708-DS; Agilent Technologies, Santa Clara, CA, USA) at 100 rpm in 900 mL dissolution media warmed to 37.5 °C. Next, Ω-3 formulations were filled in a size 00 hard capsule with an equivalent amount of 500 mg Ω-3 EE and attached to a sinker. Phosphate buffer (pH 6.8) with 0.25% (*w*/*v*) Triton X-100 was used as the dissolution medium to simulate GIT conditions. No replacement with fresh dissolution media was performed. During the experiment, temperature of the medium was maintained at 37 °C, and a sinker was used to prevent the capsule from floating. Five-milliliter aliquots of the medium solution were obtained at 5, 10, 15, 30, 45, 60, 90, and 120 min and filtered through a 0.45 µm PTFE sterile syringe filter (Pall Corporation). At this time, medium supplementation was not performed. Quantification of Ω-3 EE in the samples taken at each time point was performed through HPLC analysis, as described in Section 2.8. Omacor soft capsules, a product containing 1000 mg of Ω-3 EE, were used as the reference.

### 2.6. In Vitro Permeation

The in vitro permeation test was carried out in side-by-side cells (H6-02; Perme Gear, Hellertown, PA, USA), as described previously [73]. The hydrophilic polypropylene membrane was coated with 20% (*w*/*w*) lecithin, dissolved in *n*-dodecane, and incubated for 20 min before adding the donor/acceptor medium. Thereafter, the coated membrane was inserted between the side-by-side cell and 5 mL of the donor/acceptor medium. FeSSIF (pH 5.0) with 0.005% Triton X-100 was then added to the donor and acceptor chambers. Subsequently, the Ω-3 formulations were placed in a donor chamber, and 1 mL aliquots were withdrawn from the acceptor chamber at the scheduled time points, i.e., 0.5, 1, 2, 3, 4, 5, and 6 h. The withdrawn volumes were replaced with a fresh donor/acceptor medium. EPA and DHA were quantified, as described in Section 2.8.

### 2.7. Pharmacokinetics Study

To determine the improvement in oral absorption by SALCs, a pharmacokinetic test of the F6 formulation, which showed excellent dissolution and permeability among all the formulations and Omacor soft capsules (used as the reference), was performed using beagle dogs.

Male beagle dogs (2–3 years old, bodyweight 9–14 kg) were randomly divided into two groups (*n* = 5 per group) and acclimatized at constant temperature and humidity and ad libitum feeding for 1 week before the experiment. The dogs were fasted for at least 16 h before beginning the experiments and kept until the end of the experiments. The F6 formulation and reference were orally administered with 25 mL of water. The dose of Ω-3 EE was expressed as the amount of total EPA/DHA administered (1000 mg/head). Approximately 3 mL of blood samples from the jugular vein was collected in 10 μL heparin-treated EP tubes at 0, 1, 2, 2.5, 3, 4, 5, 6, 7, 8, 10, 12, and 24 h after administration of the formulations to beagle dogs for 24 h and immediately centrifuged (4 °C, 4000 rpm for 10 min) to separate the plasma and blood cells. Blood samples were stored at −70 °C until the EPA and DHA assays, which were performed using HPLC-MS/MS. Chromatographic separation was conducted using a Luna C18 (2) 3 μm (150 mm × 3.0 mm) column (Phenomenex. Torrance, CA, USA) equipped with an AJ0-8828 guard column (4.0 mm × 2.0 mm; Phenomenex). An acetonitrile solution, acetonitrile:water (85:15, *v*/*v*), was used as the mobile phase. The column oven was maintained at 40 °C, the flow rate was set at 0.3 mL/min with isocratic elution, and the injection volume was 5 μL. DHA-d5 was used as the internal standard. Electrospray ionization (ESI) negative mode for EPA, DHA, and DHA-d 5 was used for the analysis. The temperature of the turbo ion spray and ion spray voltages for the negative mode on ESI were set at a dwell time of 100 ms: fragmentor voltage = 135 V; collision energy = 2 eV; gas temperature = 350 °C; gas flow = 10 L/min; nebulizer = 35 psi. Detection of the ions was carried out in the multiple reaction monitoring modes (MRM) by monitoring the transition pairs of *m*/*z* 301.3 precursor ions to the *m*/*z* 257.2 productions for EPA, *m*/*z* 327.3 precursor ions to the *m*/*z* 283.4 productions for DHA [39], and *m*/*z* 332.4 precursor ions to the *m*/*z* 288.4 DHA-d 5 for the internal standard.

The experimental animal procedures were authorized by the Animal Welfare Act and guidelines for the care and use of laboratory animals by the ethics committee of the KPC Lab in Gwangju, Korea, approved on 31 July 2019 (IRB No. P192021; Identification code: E2019192).

### 2.8. HPLC Analysis

The raw material (500 mg) was precisely weighed, placed in a 100 mL volumetric flask, and marked with methyl alcohol (5 mg/mL). This solution was referred to as the sample stock solution. A total of 1, 1, 5, and 1 mL of the stock solution was placed in 100, 50, 50, and 5 mL volumetric flasks, respectively, and marked with methyl alcohol (50, 100, 500, and 1000 ppm, respectively). This solution was the standard solution. Five-milliliter aliquots of the standard solutions were filtered using 0.45 μm PTFE and placed in an HPLC vial. In addition, approximately 5 mL of the sample solution was filtered using 0.45 μm PTFE and placed in an HPLC vial. The HPLC device used was a 1260 Infinity HPLC system (Agilent Technologies). In addition, an octadecyl silyl group (C18) column (Luna, Phenomenex) with a 150 mm × 4.6 mm guard column and 5 μm pore size was used. An isocratic mobile phase consisting of acetonitrile/water at a ratio of 90/10 (*v*/*v*) was used at a flow rate of 1 mL/min. The detection wavelength using a UV spectrometer was 215 nm, and the injection volume for each sample was 10 μL.

### 2.9. Statistical Analysis

All data were expressed as mean ± standard deviation and analyzed using Student’s *t*-test to compare the means of the two groups, or analysis of variance (ANOVA) to evaluate the impact of one or more factors, including the formulation, period, and sequence of crossbar layout for administration, on the pharmacokinetic parameters as fixed effects and a random selection effect of subjects nested within the sequence [43]. Statistical significance was set at *p* < 0.05.

## 3. Results and Discussion

### 3.1. Ternary Phase Diagram

The liquid lipid crystals formed in various structures, including lamellar, cubic, and hexagonal phases, and were composed of fundamental liquid crystal-forming agents and auxiliary components, depending on their constituents and compositions. Phospholipids were selected as vital components to form liquid crystals for establishing SALCs with Ω-3 EE. Oleic acid, an unsaturated fatty acid, was added to the Ω-3/phospholipid system as an additional ingredient to induce non-lamellar phase formation, as well as to solubilize phospholipids. Ternary phase diagrams of Ω-3 EE, OA, and phospholipids in the GI fluid are shown in Figure 1. Each marker letter represents the following: the ‘N’-marked region indicates the area where the three-phase mixture was not completely mixed (phospholipids were not dissolved); the ‘L’-marked region indicates a lamellar structure; the ‘H’-marked region indicates the hexagonal phase; the ‘Q’-marked region indicates the cubic phase; the ‘B’-marked region indicates the area where two structures (lamellar and cubic, cubic, and hexagonal) overlapped.

As shown in Figure 1, the lamellar structures of the lipid phase appeared in the form of oily streaks and Maltese crosses when observed under a polarizing microscope. Furthermore, cubic structures appeared on a dark background, whereas the hexagonal structures showed a birefringence fan-like texture.

When the proportion of phospholipids was over 70%, the three components were immiscible, and no liquid crystal structure was observed in these compositions. The heterogeneous area was not determined because it was impossible to measure it using polarization microscopy. When the proportion of phospholipids was 55–60%, lamellar structures, typically a viscous liquid crystalline phase among liquid crystals, were observed; these comprised a semi-solid phase with a high viscosity. If the viscosity of the mixture is high in terms of product production, it may cause problems while filling the soft capsules after mixing the contents. When the ratio of phospholipids was 50% or less, lamellar or cubic structures were observed, depending on the proportion of Ω-3 fatty acids and OAs; mostly cubic structures were observed.

In the region where the ratio of phospholipids was less than 50%, lamellar structures were mainly observed when the *w/w* ratio of Ω-3 EE to OAs was 9:1. In contrast, in the region where the ratio of phospholipids was approximately 40–50%, a hexagonal structure was observed when the *w/w* ratio of Ω-3 fatty acids to OAs was in the range of 7:3–9:1. In the area where the phospholipid ratio was less than 30%, Ω-3 EE and OA cubic structures were predominantly formed in the region where the *w/w* ratio of Ω-3 EE to OA was approximately 8:2–9:1. The mixing ratio of Ω-3 fatty acids, phospholipids, and OA affected the formation of liquid crystals with lamellar and non-lamellar structures (hexagonal or cubic structure). The relationship between liquid crystals and the structure of constituents was also confirmed through the critical packing parameter (CPP), which could predict the molecular behavior of the amphipathic material. On the one hand, the CPP value of phospholipids is approximately 1; thus, they are predominantly known to form lamellar structures in the aqueous phase. On the other hand, OAs with a free carboxyl group (free -COOH) that can be ionized according to the pH of the solution have a CPP > 1; similarly, Ω-3 fatty acids with low ratios of hydrophilic to hydrophobic regions also have CPP > 1. Additives with a CPP > 1 have been reported to cause a phase transition from the lamellar phase towards the hexagonal or cubic phase [74]. Mixtures with a CPP approximately equal to 1 have been reported to form a sheet-like bi-layer structure, while blends with a CPP > 1 exist as hexagonal or cubic structures in a solution. A mixture of Ω-3 fatty acids and OAs was mixed with phospholipids to form a liquid crystal structure, and the resultant structure was hexagonal or cubic, but not lamellar. Non-lamellar liquid crystalline shapes, such as cubic, have a larger surface area of the lipid/water interfaces correlated with the lamellar structure; thus, their nanoparticles have excellent solubilization for Ω-3 fatty acids, which has to be predicted at a high absorption rate.

### 3.2. Characterization of the Formulation

The F1 formulation had the same composition as that of a general commercial preparation, and it did not form liquid crystals, as its composition did not contain an additive such as a liquid crystal former; F2 showed a liquid crystal with a lamellar structure shape. F3 showed a liquid crystal with a hexagonal structure, and F4, F5, and F6 showed a cubic liquid crystal configuration, as shown in Figure 1b.

### 3.3. Effects of Liquid Crystalline Phase on Particle Size and Morphology

Table 2 and Figure 2 show the results of DLS analysis. The F1–F6 formulations revealed a GI tract-like test solution. In the F2 formulation, a lamellar structure formed relatively large microparticles of 668.8 ± 44.1 nm in size. The hexagonal structure of F3 developed particles with a size of 447.5 ± 141.4 nm. The F4 formulation with a 3:7 ratio of Ω-3 fatty acids and phospholipids/OA as the main component developed particles with a size of 421.5 ± 210.7 nm in a cubic structure. On the contrary, the formulations with Ω-3 fatty acid and phospholipid/OA ratios of 6:4 (F5) and 7:3 (F6) spontaneously formed cubic nano-sized particles of 287.9 ± 189.7 and 237.5 ± 151.0 nm, respectively.

In recent years, in vitro lipolysis models such as FeSSIF have provided excellent simulations of the lipid digestion process in vivo. FeSSIF models have been used to evaluate lipid-based delivery systems to improve our understanding of drug solubilization and phase behavior. FeSSIF composed of bile salts and lecithin reportedly reduced the particle size of the formed SALCs but did not induce phase transition. Other test solutions containing enzymes such as lipase may cause phase transition. In the GI tract-like test solution prepared with FeSSIF (pH 5.0) and formulated to resemble the GI tract environment, the Ω-3 EE formulation alone or the lamellar structure (P-lamellar) formulation formed relatively large microparticles of 600 nm or more. In the case of the hexagonal or cubic structure (P-hexagonal and P-cubic) formulations, nanoparticles of 400 nm size spontaneously developed in the GIT. Therefore, it is predicted that smaller nanoparticles will be formed in an environment accompanied by external forces, such as peristalsis in the actual GI tract.

The morphology of the F6 formulation was evaluated for an in-depth study of the improvement in solubilization and absorption rates of Ω-3 fatty acids and the structural correlation of liquid crystals. Figure 3 shows the evaluation results of the structural properties (FE-TEM and AFM) of SALCs. It was confirmed that cubosomes with a cubic internal structure of approximately 250 nm or less were formed. When observed under a polarized light microscope, the cubic phase was observed as a black or dark background, as this phase is optically isotropic. The hexagonal phase exhibited a fan-like or angular texture, while the lamellar phase showed an oily streak or Maltese cross texture [75]. Additionally, the FE-TEM images revealed cubosomes that were nearly spherical with rectangular shapes, consistent with the literature [76,77]. To analyze the internal structure of the cubosome more accurately, an evaluation using cryo-TEM that does not affect the internal structure is required. Nevertheless, since cubosomes, hexosomes, and liposomes are morphologically distinct, reliable data can still be obtained using conventional TEM [78]. AFM revealed a typical cubosome structure with the corners of the squares at an angle of approximately 90° and a major portion of clearly faceted cubic structure, oriented with one side parallel to the mica surface, probably as a consequence of the drag experienced by the cubosomes during tip scanning [79,80].

The particle sizes measured in TEM and AFM were approximately 255 nm and 275 nm, respectively, and the particle sizes measured via DLS were approximately 237.5 ± 151 nm. The lager particle size measured with AFM compared with that measured via DLS due to the flattening effect of cubosomes on the Si wafer [79,80]. The mesoporous liquid crystalline architecture and membranous compartmental structure were connected. Size distribution of cubosome measured via DLS confirmed that the dispersions were monodisperse without aggregation.

### 3.4. In Vitro Permeation

Figure 4 shows results of the in vitro permeation test using side-by-side cells. When the integrity of the membrane coated with 20% (*w*/*w*) lecithin in *n*-dodecane was evaluated using methylene blue, a hydrophilic material, the membrane used in the in vitro permeation test was found to be suitable owing to its remarkably low transmittance. The apparent permeation coefficients (P_app_) of eicosapentaenoic acid (EPA) and docosahexaenoic acid (DHA) are shown in Table 3. The composition forming the liquid crystal structure (F2–F6) showed enhanced membrane permeability compared with that of the non-lipid crystal structure Ω-3 EE formulation (F1). When the permeability of the artificial membrane was compared among SALCs, it was found to increase in the following order: lamellar structure (F2) < hexagonal structure (F3) < cubic structure (F4, F5, F6). Since the cubic structure spontaneously formed nanoparticles of the smallest size in the GI tract-like test solution and membrane permeability was highest, it showed a higher increase in solubility and permeability than the other lipid structures. The P_app_ for EPA and DHA was higher in F6 than in any other formulation. The membrane permeability increased from 11% to 509% in terms of the EPA component and from 21% to 841% in DHA of Ω-3 EE. The non-lamellar structure (hexagonal and cubic) formulations contain relatively small particles owing to the high solubilization effect on Ω-3 fatty acids. The permeation of the drug can be optimized by reducing the particle size or by using an absorption enhancer [81]. The oral absorption of drugs is affected by drug solubility and permeability in the intestinal tract, the two dominant aspects affecting oral drug absorption, particularly in poorly soluble compounds [82]. After solubilization of the main ingredient into the cubic structure, i.e., cubosome, Ω-3 EE showed increased permeation into the membrane following the formation of liquid crystal particles [81,82].

### 3.5. In Vitro Dissolution

The results for EPA dissolution of the F1–F6 formulations in the biomimetic intestinal solution are shown in Figure 5a. The cumulative dissolution rates of the EPA from F1 Ω-3 fatty acids and lamellar F2 over 60 min were 10.64 ± 1.93% and 20.20 ± 4.60%, respectively. The water-immiscible nature of Ω-3 fatty acids resulted in a low F1 dissolution rate in the solution. As assessed from DLS measurement, F2 of the lamellar structure formed particles with a relatively large size (668.8 ± 44.1 nm) compared with the sizes of the particles of other formulations (F3, F4, F5, and F6) with non-lamellar structures (hexagonal or cubic structure). The lamellar structure with a relatively small surface area had a lower solubilizing effect on Ω-3 fatty acids than the non-lamellar structure. The EPA from F3, which formed a hexagonal structure, showed a dissolution rate of 42.31 ± 6.79% at 60 min. The dissolution rates of the EPA from F4, F5, and F6, which formed cubic SALC structures, were 50.48 ± 0.72%, 73.71 ± 7.25%, and 91.92 ± 0.74% at 60 min, respectively.

Figure 5b shows the DHA dissolution results for F1–F6 formulations in the biomimetic test solution. The dissolution rate of the DHA from F1, an Ω-3 fatty acid, was 10.23 ± 1.87%, while the dissolution rates of the DHA from F2 (lamellar structure) and F3 (hexagonal structure) were 14.80 ± 3.4% and 37.57 ± 6.74%, respectively. The cumulative dissolution rate of SALCs forming a cubic structure with Ω-3 fatty acids and phospholipid/OAs was 44.16 ± 0.88% in the F4 formulation, 73.02 ± 10.94% in the F5 formulation, and 90.50 ± 1.33% in the F6 formulation at 60 min, exhibiting a similar trend to that of cumulative EPA dissolution rate at 60 min.

These findings suggested that the dissolution rates of the EPA from F5 and F6 were 1.4 to 1.8-fold higher than the dissolution rate of the EPA from F4, respectively. Furthermore, the results indicated that the dissolution rates of the DHA from F5 and F6 were 1.6 to 2.0-fold higher than the dissolution rate of the DHA from F4, respectively. The dissolution rates of the EPA and DHA from F5 and F6, both with a cubic structure, were significantly higher than those in the other formulations. The dissolution rate was highest in F6, wherein the ratio of the main component and phospholipid/OA was 2:1 (*w*/*w*), implying that the improvement in the in vitro dissolution of Ω-3 fatty acids was greater in the cubic structure than in the lamellar and hexagonal structures. As the cubic structure spontaneously formed nanoparticles of the smallest size in the GI tract-like test solution, this structure showed greater improvement in solubility and dissolution than the other lipid structures. Nanocarriers formed in the GI tract increased the passage of unstirred water layer and the solubilization of Ω-3 fatty acids. These results together with in vitro permeation profiles indicate that SALCs improve solubility and permeability of Ω-3 fatty acids by forming nanoparticles similar to digestion caused by bile acids and phospholipids in the body after a meal. In terms of the in vitro dissolution test and in vitro permeation, F6 (cubic structure formulations) showed the best cumulative dissolution release and P_app_.

### 3.6. Pharmacokinetic Study

A summary of the pharmacokinetic data obtained following oral administration of the F6 formulation (1000 mg) and Omacor soft capsules (1000 mg) to beagle dogs is shown in Table 4. The plasma concentrations of EPA from the F6 formulation and commercially available Omacor soft capsules in beagle dogs are shown in Figure 6a, and the plasma concentration of DHA is shown in Figure 6b. Results revealed that the *T_max_* (h)_,_ C_max_ (ng/mL), and AUC_all_ (ng·h/mL) of EPA were 2.10 ± 0.52 h, 2051.62± 698.10 ng/mL, and 7463.52 ± 2970.58 ng·h/mL, respectively, in the reference group; and 2.15 ± 0.53 h, 5552.39 ± 3303.21 ng/mL, and 18,196.18 ± 9161.65 ng·h/mL, respectively, in the test group. Similarly, the *T_max_*, C_max_ (μg/mL), and AUC_all_ of DHA were 2.00 ± 0.62 h, 2308.63 ± 937.07 ng/mL, and 7130.45 ± 3421.18 ng·h/mL, respectively, in the reference group; and 2.15 ± 0.53 h, 7728.04 ± 4649.84 ng/mL, and 22,315.91 ± 13,207.43 ng·h/mL in the test group, respectively.

The C_max_ (µg/mL) and AUC_all_ (µg·h/mL) of EPA for the F6 formulation and SALCs were approximately 2.5-fold higher than those for the reference (Omacor soft capsules) sample. Similarly, the C_max_ (μg/mL) and AUC_all_ (μg·h/mL) of DHA for the test sample were 7728.04 ± 4649.84 ng/mL and 22,315.91 ± 13,207.43 ng/mL, which was approximately 3.1-fold higher than the 2308.63 ± 937.07 ng/mL and 7130.45 ± 3421.18 ng·h/mL for the reference sample, respectively. These results showed that SALC formulations successfully enhanced the bioavailability of Ω-3 EE because of the improvement in dissolution and permeation, resulting in the formation of a cubic structure, which spontaneously formed nanoparticles of the smallest size in the GI tract and improved the solubility and dissolution of Ω-3 EE (Figure 7). When SALCs are formed spontaneously, they can be further solubilized and degraded by the bile salts and lipolytic enzymes in the small intestine [83,84]. Therefore, the absorption of Ω-3 EE was further increased.

## 4. Conclusions

In this study, we prepared and optimized self-assembled liquid crystalline nanoparticles for enhancing the absorption of Ω-3 EE using phospholipids and OA. Briefly, Ω-3 formulations were prepared by mixing Ω-3 EE, OA, and phospholipids in a specific ratio. The ternary phase diagram was designed to determine the region of liquid crystalline structure and the effect of the composition of Ω-3 EE, OA, and phospholipids on the liquid crystalline nanoparticle properties. Morphological studies were used to characterize the stable crystal properties, while liquid crystal size analysis was conducted using polarized optical microscopy, FE-TEM, and AFM. Furthermore, a comparative study of the in vitro release patterns of EPA and DHA was performed for the six different formulations. The SALC formulations that formed a cubic structure significantly improved the dissolution and permeability of Ω-3 EE compared with those that formed hexagonal and lamellar structures or non-liquid crystalline structures. This phenomenon may be attributed to the cubic structure, which has a larger surface area and inner volume than other liquid crystalline systems and confers an excellent solubilization ability. The pharmacokinetics test revealed an optimized Ω-3 EE formulation comprising phospholipids and OAs for enhancing absorption. Results showed that the dissolution rates of the EPA and DHA from Ω-3 SALCs were 2.3 and 1.7-fold higher, respectively, than the dissolution rate of Omacor soft capsules. Moreover, an in vivo pharmacokinetic study of Ω-3 SALCs in male beagle dogs demonstrated that Ω-3-SALC oral bioavailability increased 2.5-fold for EPA and 3.1-fold for DHA, relative to the reference product. Considering the continuous secretion of bile acids from the GI tract of rats, the present study has a limited understanding of the preprandial situation.

It was established that Ω-3 EE, phospholipids, and OAs could spontaneously form liquid crystalline nanoparticles in the GI tract after oral administration. The liquid crystal structures appeared in the form of cubic, hexagonal, and lamellar structures and were determined by the composition ratio of their components. Liquid crystals exhibiting a cubic structure showed excellent solubilization ability and improved oral absorption of poorly soluble drugs. Overall, the results of this study provide useful information for improving the solubilization and absorption of poorly soluble drugs in the future. The novel approach for creating self-assembled liquid crystalline nanoparticles presented here is among the attempts to improve patient compliance when they are administered excess excipients that increase the oral absorption of oil-soluble drugs, such as Ω-3 fatty acids.

Our findings offer insights into the impact of nanoparticle phase on the oral delivery of oil-soluble drugs and provide a novel formulation of Ω-3 EE that improves the bioavailability of EPA and DHA. Further studies will be conducted in animal models of various species to identify several factors that may affect the absorption of Ω-3 fatty acids, such as food effects.

## Figures and Tables

**Figure 1 pharmaceutics-14-00068-f001:**
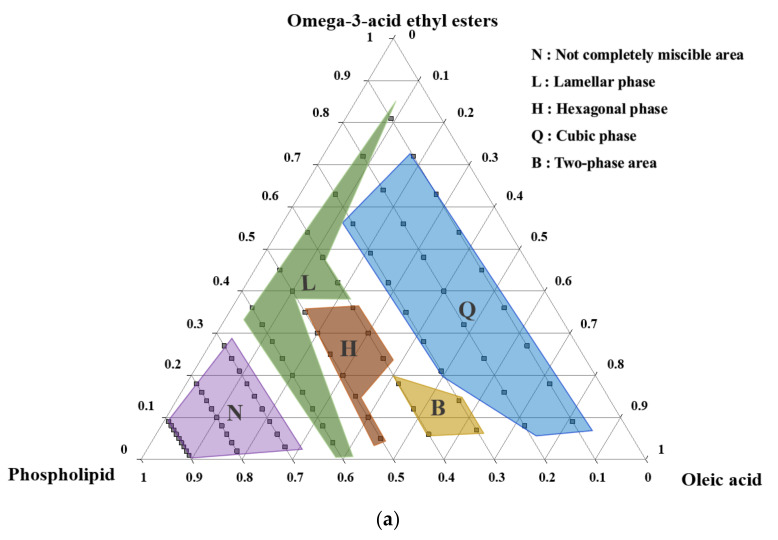
(**a**) Ternary phase diagram of Ω-3-acid ethyl esters (Ω-3 EE), oleic acid, and phospholipids in water at 25 °C. (**b**) Lamellar, (**c**) hexagonal, and (**d**) cubic structures observed via polarized optical microscopy, showing the morphology of liquid crystalline nanoparticles.

**Figure 2 pharmaceutics-14-00068-f002:**
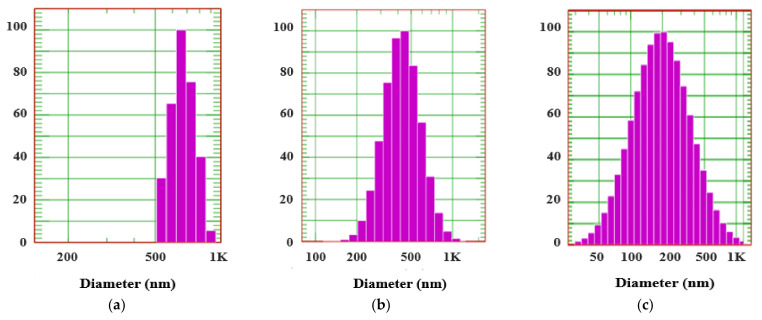
Particle size distribution of the self-assembled liquid crystalline nanoparticles; (**a**) F2 formulation (lamellar structure), (**b**) F3 formulation (hexagonal structure), (**c**) F6 formulation (cubic structure).

**Figure 3 pharmaceutics-14-00068-f003:**
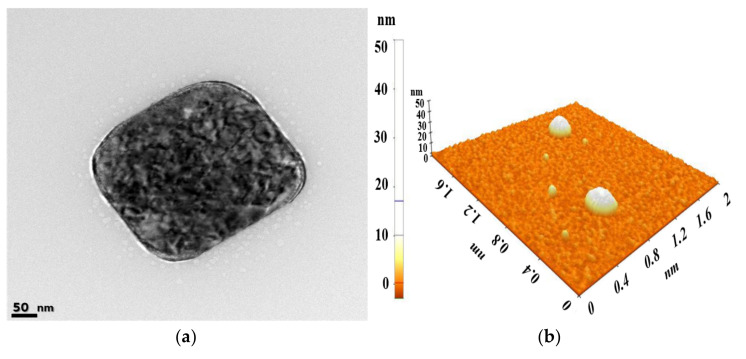
Morphological characterization of PL-SALCs (F6 formulation) using (**a**) FE-TEM (scale bar represents 50 nm) and (**b**) AFM. Abbreviations: SALCs, self-assembled liquid crystalline nanoparticles; FE-TEM, field emission transmission electron microscopy; AFM, atomic force microscope.

**Figure 4 pharmaceutics-14-00068-f004:**
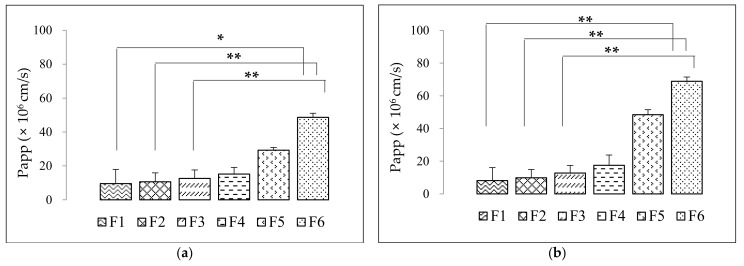
Apparent permeation coefficient (P_app_) for (**a**) eicosapentaenoic acid (EPA) and (**b**) docosahexaenoic acid (DHA) of F1–F6 formulations in the in vitro permeation test. * *p* < 0.05 is established between SALC formulations. Data are presented as mean ± SD (*n* = 3). ** *p* < 0.01 is established between SALC formulations. Data are presented as mean ± SD (*n* = 3).

**Figure 5 pharmaceutics-14-00068-f005:**
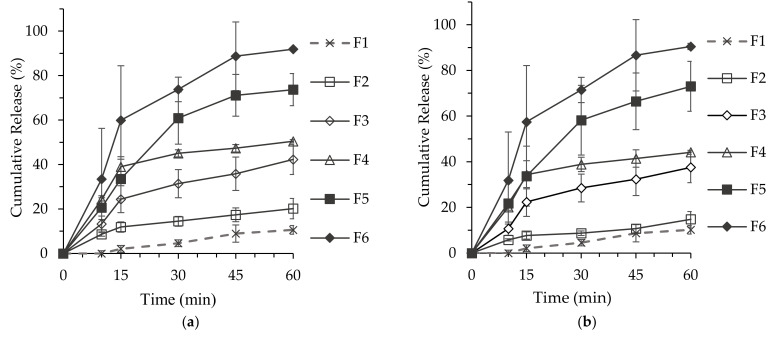
Dissolution of Ω-3 fatty acids: oleic acid: phospholipids vs Ω-3 acid ethyl esters. (*n* = 3), (**a**) In vitro dissolution rate of the EPA (eicosapentaenoic acid) from F1–F6; (**b**) In vitro dissolution rate of the DHA (docosahexaenoic acid) from F1–F6.

**Figure 6 pharmaceutics-14-00068-f006:**
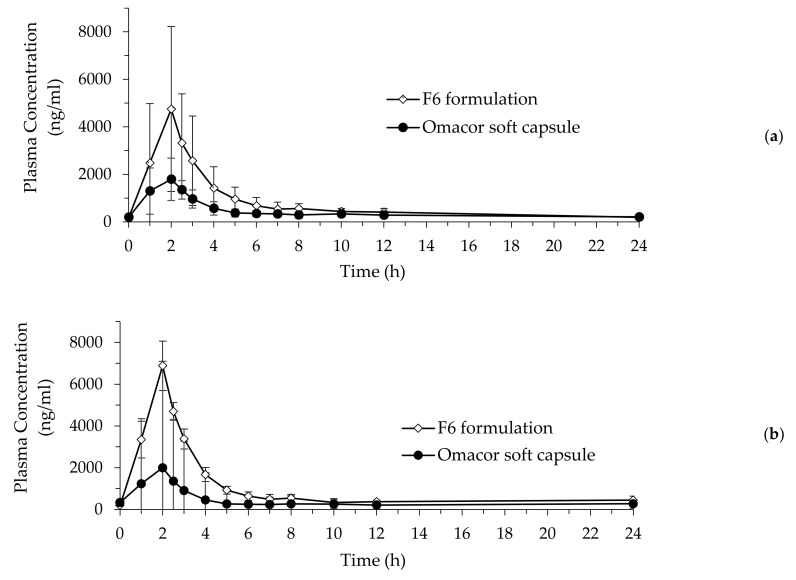
Plasma concentration time profile of SALCs (F6 formulation) and the reference (Omacor soft capsules) after oral administration of a 1000 mg single-dose of Ω-3 EE in male beagle dogs (*n* = 5). (**a**) Plasma concentration of EPA in SALCs and the reference; (**b**) Plasma concentration of DHA in SALCs and the reference. Abbreviations: EPA, eicosapentaenoic acid; DHA, docosahexaenoic acid.

**Figure 7 pharmaceutics-14-00068-f007:**
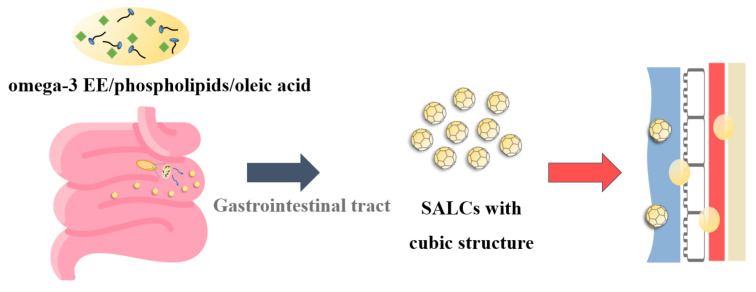
Schematic showing the mechanism of action that promotes omega absorption in SALC formulations. Abbreviations: EPA, eicosapentaenoic acid; DHA, docosahexaenoic acid.

**Table 1 pharmaceutics-14-00068-t001:** SALC formulations (weight %/unit).

Ingredient	F1	F2	F3	F4	F5	F6
Ω-3 EE	100	30	30	30	55.6	66.6
SPC	-	10	30	50	22.2	16.7
OA	-	60	40	20	22.2	16.7
Total	100	100	100	100	100	100

Abbreviations: Ω-3 EE, Ω-3 ethyl ester; SPC, soybean phosphatidylcholine; OA, oleic acid.

**Table 2 pharmaceutics-14-00068-t002:** Particle size distribution of the self-assembled liquid crystalline nanoparticles.

Gaussian Intensity (nm)	F1	F2	F3	F4	F5	F6
Mean ± SD	973.7 ± 360.3	668.8 ± 44.1	447.5 ± 141.4	421.5 ± 210.7	287.9 ± 189.7	237.5 ± 151

**Table 3 pharmaceutics-14-00068-t003:** Apparent permeation coefficient of F1–F6 formulations in vitro permeation test.

P_app_ (× 10^6^) (cm/s)	F1	F2	F3	F4	F5	F6
EPA (Mean ± SD)	9.54 ± 8.38 ^a^	10.64 ± 5.01 ^b^	12.61 ± 5.22 ^b^	15.17 ± 3.94	29.28 ± 1.60	48.64 ± 2.40 ^a,b^
DHA (Mean ± SD)	8.13 ± 7.95 ^b^	9.84 ± 4.72 ^b^	12.73 ± 5.09 ^b^	17.49 ± 6.28	48.40 ± 3.09	68.90 ± 2.62 ^b^

^a^ *p* < 0.05 is established between SALC formulations. Data are represented as mean ± SD (*n* = 3). ^b^ *p* < 0.01 is established between SALC formulations. Data are represented as mean ± SD (*n* = 3).

**Table 4 pharmaceutics-14-00068-t004:** Plasma concentration of the SALCs (F6) and reference (Omacor) after oral administration with a single dose of 1000 mg/head. Each value represents arithmetic mean ± SD (*n* = 5).

	EPA (Mean ± SD)		DHA (Mean ± SD)	
	Omacor Capsule	F6 Formulation	Omacor Capsule	F6 Formulation
*T_1/2_* (h)	11.77 ± 7.57	10.89 ± 3.14	14.69 ± 10.78	15.82 ± 6.81
*T_max_* (h)	2.1 ± 0.52	2.15 ± 0.53	2.0 ± 0.62	2.15 ± 0.53
C_max_ (ng/mL)	2051.62 ± 698.10	5552.39 ± 3303.21	2308.63 ± 937.07	7728.04 ± 4649.84
AUC_all_ (ng·h/mL)	7463.52 ± 2970.58	18,196.18 ± 9161.65	7130.45 ± 3421.18	22,315.91 ± 13,207.43
AUC_INF_ (ng·h/mL)	11,720.69 ± 5388.02	21,883.61 ± 10,133.70	12,936.85 ± 10,374.62	31,588.8 ± 19,869.53
MRT_INF_ (h)	14.81 ± 9.79	11 ± 2.61	19.98 ± 15.48	16.19 ± 6.21

Abbreviations: SALCs, self-assembled liquid crystalline nanoparticles; SD, standard deviation; *T_1/2_*, half-life; *T_max_*, time at maximum plasma concentration; C_max_, maximum plasma concentration; AUC, area under the curve; MRT, mean residence time.

## Data Availability

The data presented in this study are available upon request from the corresponding author.

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
