# Peer review of "Formation of Self-Assembled Liquid Crystalline Nanoparticles and Absorption Enhancement of Ω-3s by Phospholipids and Oleic Acids"

_pharmaceutics, 2021, doi:10.3390/pharmaceutics14010068_

Round 1
Reviewer 1 Report
The discussion about the morphology of SALC is based mainly on polarized microscopy and pictures, which are not fully convincing. I haven’t found the information about the possible influence of FeSSIF on liquid crystalline phases. The TEM image doesn’t show valuable data about the structure, and it is in the dried state which for sure affects the possible crystalline form. The main problem is that it is not straightforward presented how samples are prepared and which compositions are investigated i.e. colloidal solutions of SALC or pure SALC out of solutions? It has to be corrected. The theoretical discussion regarding CPP value is not convincing enough. The ternary system can be easily disturbed by individual components, therefore, it is hard to assume that this approach can be directly used for the simple, one-component colloidal solution. If this part is supposed to be left in the manuscript the discussion about it should to be more detailed. Another thing is that I don’t see any good evidence justifying the conclusion for the sentence that “We found that ω-3 is present in 380 the core solution of phospholipids and oleic acid”. Could you explain it? Table 2 is not clear as well as the description of fig. 2. – “Particle size distribution (intensive and weight)”. How do authors understand “intensive and weight? I also found some word repetition or not understandable paragraphs. Overall, the paper could provide interesting data, especially for the aim of ω-3s absorption enhancement, however, before publication, it has to be carefully improved in a manner of data presentation, article structure (why 3.5 Morphology paragraph is after 3.4. In vitro permeation? I think it should be together with 3.3 paragraph), and English writing (repetitions and some grammatical errors).
Author Response
Dear Reviewr 1:
We wish to resubmit the manuscript titled “Formation of Self-Assembled Liquid-Crystalline Nanoparticles and Absorption Enhancement of Ω-3s by Phospholipids and Oleic acids.” The manuscript ID is pharmaceutics-1496436.
We thank you for your thoughtful suggestions and insights. The manuscript has benefited from these insightful suggestions. I look forward to working with you to move this manuscript closer to publication in Pharmaceutics.
The manuscript has been rechecked and the necessary changes have been made in accordance with the reviewers’ suggestions. The responses to all comments have been prepared and attached herewith below.
Thank you for your consideration. I look forward to hearing from you.
Sincerely,
Young-Joon Park
College of Pharmacy, Ajou University, Worldcup-ro 206
Yeongtong-gu, Suwon-si16499
Republic of Korea
Tel: +82-031-219-3447
Fax: +82-31-219-3435
E-mail: parkyj64@gmail.com

Reviewer 2 Report
The manuscript entitled Formation of Self-Assembled Liquid-Crystalline Nanoparticles and Absorption Enhancement of Ω-3s by Phospholipids and Oleic acids” is a document of interesting subject matter.
However, it needs some major changes before being accepted. Make the following corrections:
- What is GI trace (in line 40)? All abbreviations should be defined at first appearance.
- The authors briefly mentioned the application of nanotechnology in a drug delivery In this context, authors have to site the most recent publications such as DOI: 10.3390/nano11051086; DOI: 10.1016/j.lfs.2021.119420; DOI: 10.1016/j.arabjc.2021.103321
- Check the format of Table 8. The formats should be the same in all tables. In addition, it is recommended that the table does not have vertical and minor horizontal lines. Please check Table 8.
- In line 403 Figure 5. (b) Figure 5 (b)
- The format of ºC should be the same everywhere. Please check line 165.
- In vitro or In vivo terms should be the same format everywhere. Please check them.
- In Table 3 and Figure 3, is the results checked statistically? There is a significant difference between analyses? If there is a significant difference statistically, it is recommended to mention it in the caption of the table and figure
Author Response
Dear Reviewer 2:
We wish to resubmit the manuscript titled “Formation of Self-Assembled Liquid-Crystalline Nanoparticles and Absorption Enhancement of Ω-3s by Phospholipids and Oleic acids.” The manuscript ID is pharmaceutics-1496436.
We thank you for your thoughtful suggestions and insights. The manuscript has benefited from these insightful suggestions. I look forward to working with you to move this manuscript closer to publication in Pharmaceutics.
The manuscript has been rechecked and the necessary changes have been made in accordance with the reviewers’ suggestions. The responses to all comments have been prepared and attached herewith below.
Thank you for your consideration. I look forward to hearing from you.
Sincerely,
Young-Joon Park
College of Pharmacy, Ajou University, Worldcup-ro 206
Yeongtong-gu, Suwon-si16499
Republic of Korea
Tel: +82-031-219-3447
Fax: +82-31-219-3435
E-mail: parkyj64@gmail.com

Round 2
Reviewer 1 Report
In my opinion the article is improved enough to be published.